# Shifting the PPARγ conformational ensemble toward a transcriptionally repressive state improves covalent inhibitor efficacy

Liudmyla Arifova[1,2], Brian S MacTavish[3], Zane Laughlin[2], Mithun Nag Karadi Girdhar[2], Jinsai Shang[3,4], Min-Hsuan Li[2], Xiaoyu Yu[2], Di Zhu[5], Theodore M Kamenecka[5], Douglas J Kojetin[2,3,6,7]*

[1]Undergraduate Program in Biochemistry and Chemical Biology, Vanderbilt University, Nashville, United States; [2]Department of Biochemistry, Vanderbilt University, Nashville, United States; [3]Department of Integrative Structural and Computational Biology, Scripps Research and The Herbert Wertheim UF Scripps Institute for Biomedical Innovation and Technology, Jupiter, United States; [4]School of Basic Medical Sciences, Guangzhou Laboratory, Guangzhou Medical University, Guangzhou, China; [5]Department of Molecular Medicine, Scripps Research and The Herbert Wertheim UF Scripps Institute for Biomedical Innovation and Technology, Jupiter, United States; [6]Center for Structural Biology, Vanderbilt University, Nashville, United States; [7]Vanderbilt Institute of Chemical Biology, Vanderbilt University, Nashville, United States

*For correspondence: douglas.kojetin@vanderbilt.edu

**Competing interest:** The authors declare that no competing interests exist.

## eLife Assessment

This manuscript presents a **fundamental** advance in our understanding of nuclear receptor pharmacology by expanding on previous work demonstrating dual ligand occupancy in the peroxisome proliferator-activated receptor-gamma (PPARγ). Using a **compelling** combination of biophysical, biochemical, and cellular approaches, the authors show that covalent inverse agonists with enhanced efficacy shift the receptor conformation toward a transcriptionally repressive state that limits orthosteric ligand co-binding more effectively. This revised manuscript further strengthens support for a proximal, bidirectional allosteric model of dual ligand occupancy by sharpening the distinction between prior and new findings, adding clear conceptual figures, and strengthening statistical rigor.

**Abstract** The nuclear receptor peroxisome proliferator-activated receptor gamma (PPARγ) regulates transcription in response to ligand binding at an orthosteric pocket within the ligand-binding domain (LBD). We previously showed that two covalent ligands, T0070907 and GW9662—extensively used as PPARγ inhibitors to assess off-target activity—weaken but do not completely block ligand binding via an allosteric mechanism associated with pharmacological inverse agonism (Shang and Kojetin, 2024). These covalent inhibitors shift the LBD toward a repressive conformation, where the activation function-2 (AF-2) helix 12 occupies the orthosteric pocket, competing with orthosteric ligand binding. Here, we provide additional support for this allosteric mechanism using two covalent inverse agonists, SR33065 and SR36708, which better stabilize the repressive LBD conformation and are more effective inhibitors of—but also do not completely inhibit—ligand cobinding. Furthermore, we show that ligand cobinding can occur with a previously reported PPARγ dual-site covalent inhibitor, SR16832, which appears to weaken ligand binding through a direct mechanism independent of

the allosteric mechanism. These findings underscore the complex nature of the PPARγ LBD conformational ensemble and highlight the need to develop alternative methods for designing more effective covalent inhibitors.

## Introduction

Peroxisome proliferator-activated receptor gamma (PPARγ) regulates gene expression programs that influence cellular differentiation and insulin sensitization in response to ligand binding. The discovery in 1995 that PPARγ is a molecular target for thiazolidinedione (TZD)-containing synthetic ligands (*Lehmann et al., 1995*), which were first reported in 1983 (ciglitazone from Takeda) from phenotypic screens as insulin sensitizers (*Fujita et al., 1983*), led to a significant interest in pharmacologically modulating PPARγ for therapeutic intervention in patients with type 2 diabetes. Several TZDs, including pioglitazone (Actos by Takeda), rosiglitazone (Avandia by GSK), and troglitazone (Rezulin by Daiichi Sankyo & Parke-Davis), were approved by the U.S. Food and Drug Administration (FDA) for use in patients with type 2 diabetes. However, patient reports of undesired side effects and adverse events in the 2000s associated with clinical use of TZDs that included edema, congestive heart failure, risk of bone fracture, and others led to FDA black box warnings or withdrawal from the market (*Soccio et al., 2014*).

TZDs function as pharmacological agonists that activate PPARγ-mediated transcription by stabilizing an active ligand-binding domain (LBD) activation function-2 (AF-2) coregulator surface conformation (*Shang et al., 2019*), which promotes coactivator recruitment to PPARγ-bound regions of chromatin resulting in increased expression of PPARγ target genes (*Haakonsson et al., 2013*). Recent mechanism of action studies have also revealed a potential path for separating the insulin sensitizing effects of PPARγ-binding ligands from undesired side effects. The field now appreciates that ligand binding can influence the structure and function of two distinct surfaces in the PPARγ LBD (*Frkic et al., 2021*). Ligand binding can stabilize the AF-2 coregulator interaction surface in a conformation that enhances recruitment of transcriptional coactivator complexes that regulates an adipogenic gene program or, in a non-mutually exclusive manner, inhibit obesity-linked phosphorylation of Ser273 by Cdk5 to influence expression of an insulin sensitizing gene program (*Choi et al., 2010*). Subsequent studies have focused on developing so-called next-generation selective PPARγ modulators, which contained compound scaffolds different from TZDs and comprise a wide range of pharmacological activities: PPARγ partial agonists (*Choi et al., 2010*), transcriptionally neutral antagonists (*Choi et al., 2011*), and transcriptionally repressive inverse agonists (*Marciano et al., 2015*; *Stechschulte et al., 2016*) can inhibit Cdk5-mediated phosphorylation of Ser273.

To determine if off-target effects may be responsible for the beneficial or side effects of TZDs and other PPARγ-binding compounds, researchers have used two compounds originally reported in 2002—GW9662 by GlaxoSmithKline (*Leesnitzer et al., 2002*) and T0070907 by Tularik (*Lee et al., 2002*)—as covalent inhibitor antagonists of ligand binding to the PPARγ ligand-binding domain (LBD). GW9662 and T0070907, which bind via a halogen exchange reaction to a reactive cysteine residue (Cys285 or Cys313 PPARγ isoform 1 or 2, respectively) that points into the orthosteric ligand-binding pocket of PPARγ, were shown to inhibit binding of [$^3$H]-rosiglitazone in a radiolabeled ligand-binding assay and inhibit agonist-induced transcription and adipocyte differentiation in cells cotreated with the covalent inhibitor and rosiglitazone. Crystal structures show overlapping orthosteric ligand-binding modes for covalent inhibitors (*Brust et al., 2018*; *Chandra et al., 2008*) and non-covalent PPARγ ligands including full agonists such as rosiglitazone (*Nolte et al., 1998*) or partial agonists including MRL-24 and nTZDpa (*Bruning et al., 2007*), which provided additional support for the field to use GW9662 and T0070907 as covalent inhibitors. However, in 2014, we showed that GW9662 and T0070907 do not block all ligands from binding to the PPARγ LBD—a phenomenon we originally called 'alternate site' ligand binding (*Hughes et al., 2014*). Other studies have confirmed that non-covalent synthetic ligands and cellular metabolites can cobind to the PPARγ LBD in the presence of GW9662 or T0070907 when used as covalent inhibitors (*Arifi et al., 2023*; *Brust et al., 2017*; *Hughes et al., 2016*; *Jang et al., 2017*; *Laghezza et al., 2018*; *Leijten-van de Gevel et al., 2022*; *Shang et al., 2018*).

To gain structural insight into the non-covalent and covalent ligand cobinding mechanism, we recently reported seven X-ray crystal structures of PPARγ LBD cobound to a covalent inhibitor

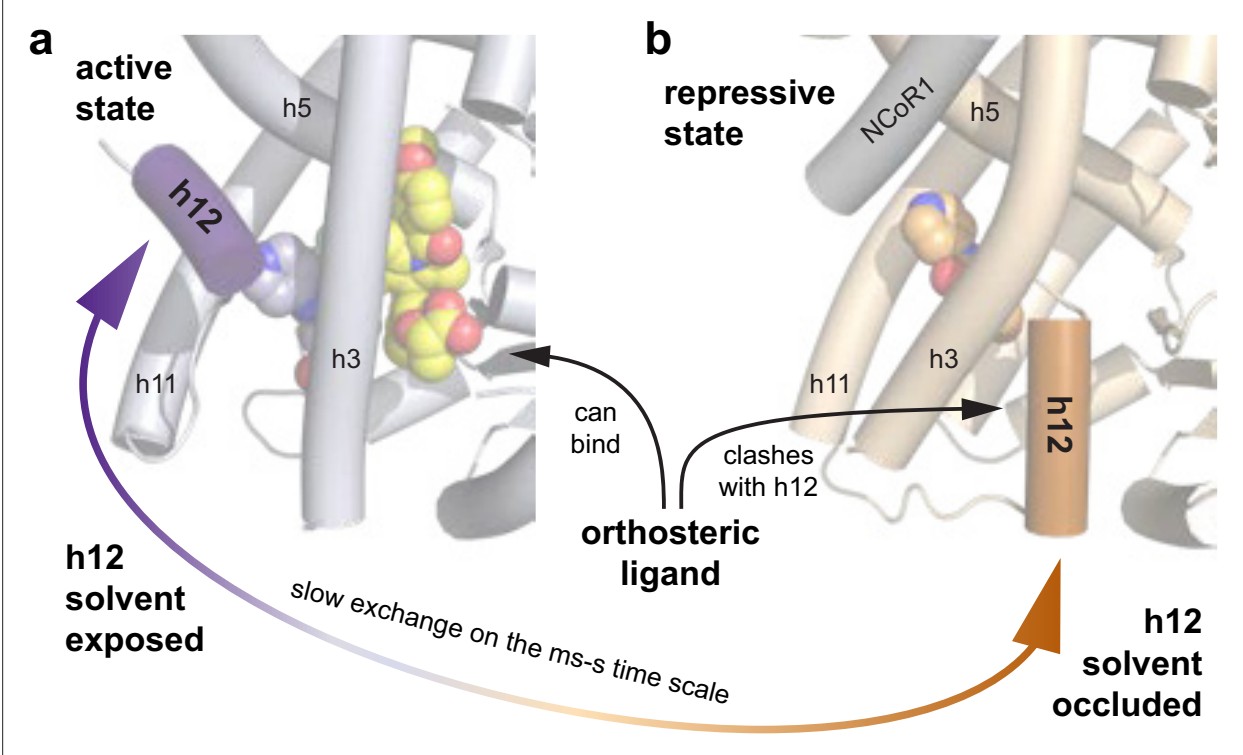

**Figure 1.** A possible allosteric mechanism by which a covalent ligand inhibits binding of other non-covalent ligands. Crystal structures of T0070907-bound PPARγ ligand-binding domain (LBD) cobound with (**a**) non-covalent agonist MRL-24 (PDB 8ZFS) and (**b**) NCoR1 corepressor peptide (PDB 6ONI). Helix 12 (h12) can adopt a solvent-exposed active conformation—or a solvent-occluded repressive conformation within the orthosteric ligand-binding pocket that would physically clash and block binding of an orthosteric ligand.

(GW9662 or T0070907) and different synthetic non-TZD PPARγ modulators (*Shang and Kojetin, 2024*). We surprisingly found that the non-covalent ligand-binding event we previously described at the alternate/allosteric site, which is proximal to the orthosteric ligand-binding pocket, can instead correspond to the ligand adopting the original orthosteric binding mode where the covalent inhibitor adopts a binding mode that permits ligand cobinding (*Figure 1a*) with some variation depending on the specific covalent and non-covalent ligand pair. Furthermore, biochemical and protein NMR studies suggested a potential mechanism explaining why T0070907, a corepressor-selective pharmacological inverse agonist, is a more effective covalent inhibitor of ligand binding than GW9662, a transcriptionally neutral pharmacological antagonist. GW9662 stabilizes an active-like LBD conformation, whereas T0070907-bound LBD exchanges between two long-lived conformations corresponding active- and repressive-like states (*Brust et al., 2018*). Crystal structures of PPARγ LBD reveal that in the transcriptionally repressive state, when cobound to T0070907 and NCoR1 corepressor peptide, a critical regulatory element called helix 12 adopts a solvent-occluded conformation within the orthosteric ligand-binding pocket (*Figure 1b*; *Shang et al., 2020*). When T0070907 is used as a covalent inhibitor, ligand cobinding resulted in the disappearance of the repressive-like state and stabilization of an active-like state that is similar to the GW9662 cobound ligand state (*Shang and Kojetin, 2024*). These data suggested a mechanism where helix 12 and a non-covalent ligand compete to occupy the orthosteric ligand-binding pocket (*Figure 1*)—and ligand cobinding in the presence of the pharmacological inverse agonist T0070907, when used as a covalent inhibitor, selects or induces an active LBD conformation where helix 12 adopts a solvent-exposed active conformation. This mechanism is consistent with the two-step mechanism described for agonist binding to PPARγ (*Shang and Kojetin, 2021*).

In the discussion of our previous study (*Shang and Kojetin, 2024*), we suggested that covalent inhibitors with improved pharmacological corepressor-selective inverse agonist functions—that better stabilize a repressive-like LBD conformation where helix 12 adopts a solvent-occluded confirmation within the orthosteric pocket—may more effectively inhibit ligand cobinding to the orthosteric pocket.

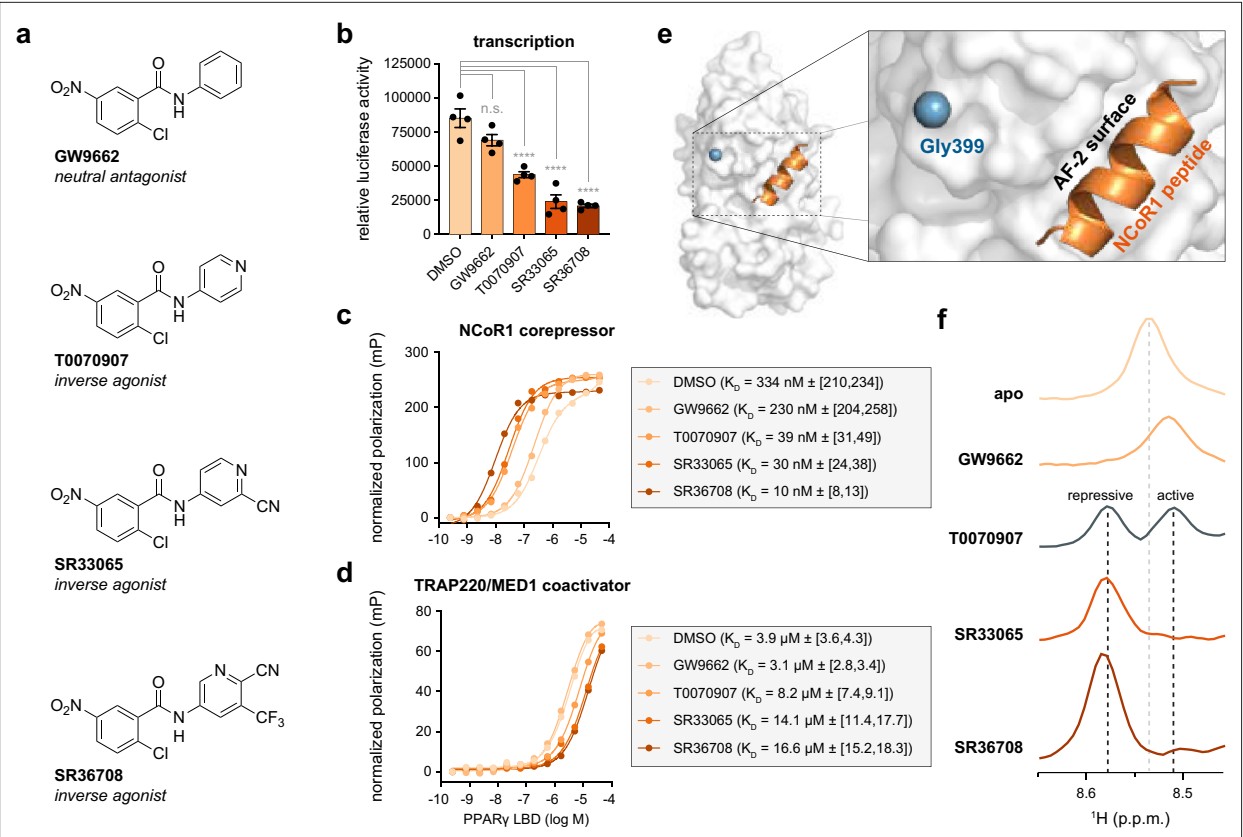

**Figure 2.** Pharmacological covalent inverse agonists stabilize a repressive ligand-binding domain (LBD) conformation. (**a**) Chemical structures of four covalent ligands with different pharmacological properties including transcriptionally neutral antagonism (GW9662) or transcriptionally repressive partial inverse agonism (T0070907) and a full inverse agonism (SR33065 and SR36708). (**b**) Cell-based luciferase reporter assay in HEK293T cells transfected with an expression plasmid encoding full-length PPARγ and a 3xPPRE-luciferase reporter plasmid (*n* = 4 technical replicates; mean ± SEM). One-way ANOVA using Dunnett's multiple comparisons test was used to compare DMSO (control) to ligand-treated conditions; ****p ≤ 0.0001. Fluorescence polarization (**c**) FITC-labeled NCoR1 corepressor and (**d**) FITC-labeled TRAP220/MED1 coactivator peptide binding assays (*n* = 3 technical replicates) with fitted affinities shown in the legend (mean ± [95% CI]). (**e**) Location of Gly399 in the crystal structure of PPARγ LBD bound to NCoR1 peptide and inverse agonist T0070907 (PDB 6ONI). (**f**) One-dimensional (1D) traces extracted from two-dimensional (2D) [$^1$H,$^{15}$N]-TROSY-HSQC NMR data of $^{15}$N-labeled PPARγ LBD in the absence or presence of the indicated covalent ligands. The gray dotted line denotes the Gly399 backbone amide chemical shift in the apo/ligand-free form; black lines denote the active and repressive chemical shift values when bound to T0070907.

Here, we test this hypothesis using two covalent inverse agonists, SR33065 and SR36708, which we recently reported with improved efficacy over T0070907 (*MacTavish et al., 2025*). Biochemical and NMR-based structural biology ligand cobinding assays show that although SR33065 and SR36708 have similar inverse agonist efficacy, they display ligand-specific differences when used as a covalent inhibitor—and they still do not completely block ligand binding to PPARγ. Finally, we also show that binding of a non-covalent ligand is not blocked by another previously described covalent inhibitor, SR16832 (*Brust et al., 2017*), which appears to function via a different mechanism that does not involve corepressor-selective inverse agonism and helix 12 occupancy of the orthosteric pocket.

## Results

### Corepressor-selective covalent inverse agonists stabilize a repressive LBD conformation

In our recent study, we reported a series of 2-chloro-5-nitrobenzamide covalent analogs that display graded PPARγ-mediated transcriptional repression from partial to full inverse agonism (*MacTavish et al., 2025*). Several compounds, including SR33065 and SR36708 (*Figure 2a*), displayed improved inverse agonist transcriptionally repressive efficacy compared to the neutral antagonist GW9662 and

partial inverse agonist T0070907. Here, we compared the activities of these compounds using cellular and biochemical ligand profiling assays. Using a cell-based luciferase reporter assay, SR33065 and SR36708 are more efficacious at repressing full-length PPARγ-mediated transcription compared to T0070907 (*Figure 2b*). Using fluorescence polarization (FP) biochemical assays with purified PPARγ LBD protein, we confirmed that improved inverse agonist efficacy associated with SR33065 and SR36708 is associated with strengthening the binding affinity of an FITC-labeled NCoR1 corepressor peptide (*Figure 2c*) and decreased binding of an FITC-labeled TRAP220/MED1 coactivator peptide (*Figure 2d*) derived from transcriptional coregulator proteins that influence ligand-dependent PPARγ transcription (*Ge et al., 2002*; *Yu et al., 2005*).

In our previous study (*MacTavish et al., 2025*), conformation–activity relationship analysis using protein NMR chemical shift perturbation (CSP) analysis showed that covalent inverse agonists with improved efficacy shift the PPARγ LBD conformational ensemble toward a repressive LBD/AF-2 surface conformation. This ligand-dependent shift in the functional LBD conformational ensemble was apparent in particular for Gly399, a residue near the AF-2 coregulator interaction surface that does not physically interact with the bound corepressor peptide (*Figure 2e*). To illustrate this, we recollected 2D [$^1$H,$^{15}$N]-TROSY-HSQC NMR data of $^{15}$N-labeled PPARγ LBD in the presence of DMSO (apo), GW9662, T0070907, SR33065, or SR36708. 1D projections extracted from the 2D NMR data focused on Gly399 (*Figure 2f*) show that SR33065 and SR36708 stabilize the AF-2 surface in a repressive conformation, GW9662 stabilizes an active-like conformation, whereas T0070907-bound PPARγ LBD populates both the repressive and active-like conformations in slow exchange on the NMR time scale resulting in the appearance of two NMR peaks. These data reaffirm our recent findings (*Laughlin et al., 2025*; *MacTavish et al., 2025*) that demonstrated covalent PPARγ inverse agonists with graded transcriptionally repressive efficacy—that is, full inverse agonism (SR33065 and SR33708) vs. partial inverse agonism (T0070907)—function via shifting the LBD conformational ensemble toward a repressive LBD conformation that increases corepressor binding affinity and decreases coactivator binding affinity.

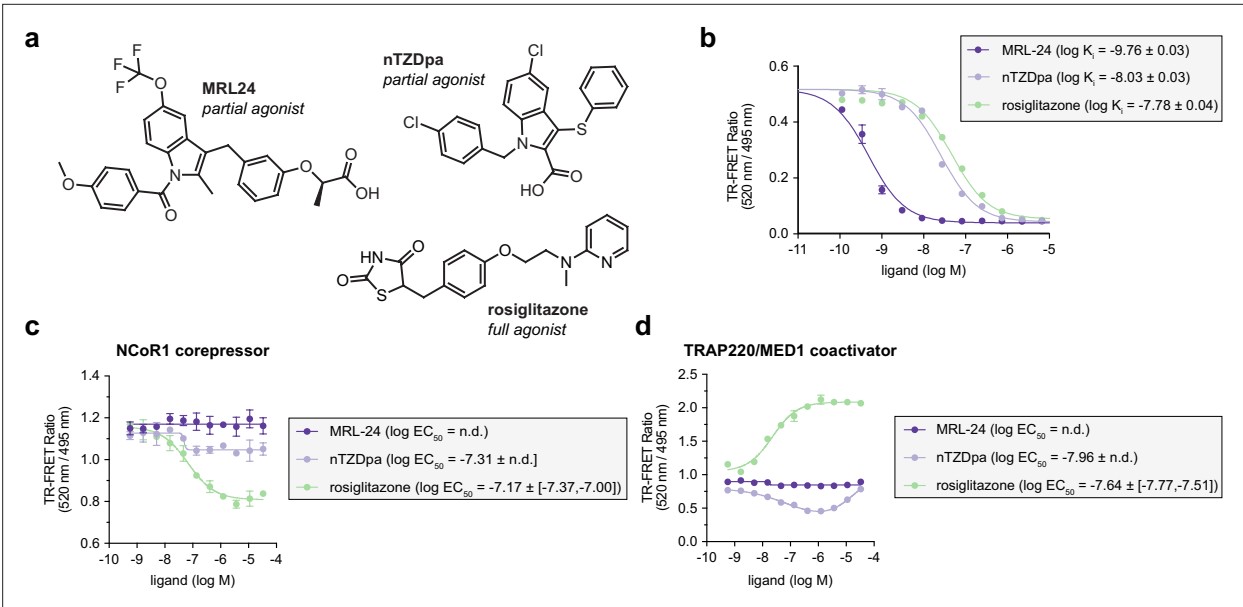

**Figure 3.** Time-resolved fluorescence resonance energy transfer (TR-FRET) profiling of non-covalent ligand binding to PPARγ ligand-binding domain (LBD). (**a**) Chemical structures of non-covalent PPARγ partial agonists MRL-24 and nTZDpa and the full agonist rosiglitazone. (**b**) TR-FRET ligand displacement assay (*n* = 3 technical replicates) with fitted $K_i$ values shown in the legend (mean ± SD). TR-FRET coregulator interaction assays where non-covalent agonists were titrated with in the presence of (**c**) 400 nM FITC-labeled NCoR1 corepressor or (**d**) 400 nM FITC-labeled TRAP220/MED1 coactivator peptide (*n* = 3 technical replicates; mean ± [95% CI]; n.d. = not determined).

## Inverse agonism and corepressor peptide binding synergize to inhibit ligand cobinding

To study ligand cobinding, we used two non-covalent PPARγ partial agonists, MRL-24 and nTZDpa (*Figure 3a*) from our previous cobinding study (*Shang and Kojetin, 2024*), which bind to the PPARγ LBD with affinities in the high picomolar and low nanomolar range, respectively (*Figure 3b*). These high-affinity ligands are partial agonists as defined on their functional outcome in coregulator recruitment and cellular transcription; that is, they are less efficacious than full agonists at recruiting peptides derived from coactivator proteins in biochemical assays (*Chrisman et al., 2018*; *Shang et al., 2019*; *Shang and Kojetin, 2024*) and increasing PPARγ-mediated transcription (*Acton et al., 2005*; *Berger et al., 2003*). Despite the high binding affinity of MRL-24, it does not show a significant concentration-dependent change in time-resolved fluorescence resonance energy transfer (TR-FRET) biochemical coregulator interaction assays using (*Figure 3c*) FITC-labeled NCoR1 corepressor or (*Figure 3d*) FITC-labeled TRAP220/MED1 coactivator peptides. In contrast, nTZDpa shows a concentration-dependent decrease in NCoR1 corepressor peptide interaction and a biphasic, bell-shaped decrease then increase in TRAP220/MED coactivator peptide interaction, consistent with previous reports that showed two equivalents of nTZDpa can bind to the PPARγ LBD with different affinities (*Chrisman et al., 2018*; *Shang and Kojetin, 2024*). These activities are somewhat distinct from the PPARγ full agonist rosiglitazone, which also binds with low nanomolar affinity, decreases NCoR1 peptide interaction, and increases TRAP220/MED1 peptide interaction.

Covalent inverse agonists strengthen NCoR1 corepressor peptide interaction (*Figure 2c*), which in TR-FRET biochemical coregulator interaction assays results in an increase in FITC-labeled NCoR1 corepressor peptide interaction TR-FRET efficacy relative to apo-PPARγ LBD (*Brust et al., 2018*; *MacTavish et al., 2025*). We previously showed that the increased corepressor-selective efficacy afforded by T0070907, resulting from occupancy of helix 12 within the orthosteric pocket leaving the AF-2 surface fully exposed to interaction with corepressor peptide (*Shang et al., 2020*), provides a sensitive window to observe alternate site ligand cobinding (*Hughes et al., 2014*; *Shang and Kojetin, 2024*).

We performed biochemical TR-FRET coregulator assays using PPARγ LBD preincubated covalent ligands with different pharmacological activities—antagonist (GW9662), partial inverse agonist (T0070907), or full inverse agonist (SR33065, SR36708)—and titrated MRL-24 (*Figure 4a*) or nTZDpa (*Figure 4b*) to determine $IC_{50}$ potency values for displacing an FITC-labeled NCoR1 corepressor peptide. Furthermore, we performed these assays using increasing concentrations of the FITC-NCoR1 peptide to determine how synergistic forcing the LBD into a repressive conformation along with increasing corepressor-selective inverse agonist efficacy influences the ligand cobinding inhibitory functions of the covalent ligands. TR-FRET data plotted TR-FRET ratio vs. non-covalent ligand concentration show that increasing FITC-NCoR1 peptide concentrations in the assay leads to higher TR-FRET ratio values indicating higher occupancy of the peptide-bound state. Furthermore, PPARγ LBD bound to more efficacious covalent inverse agonists, SR33065 and SR33708, also displays higher occupancy of the peptide-bound state as TR-FRET values under increasing FITC-NCoR1 peptide concentrations are higher than that of T0070907 and GW9662. When plotted as normalized TR-FRET ratio vs. non-covalent ligand concentration, the data indicate that SR33065 and SR33708 weaken cobinding of MRL-24 more than T0070907 and GW9662. In general, the data also show that the covalent ligands weaken nTZDpa cobinding, though the results are more nuanced.

To more quantitatively analyze the data, we compared parameters reported from a fit of the TR-FRET concentration response curves as a function of covalent ligand identity and FITC-labeled NCoR1 corepressor peptide concentration used in the assay using a four-parameter dose–response equation. Fitted parameters from this analysis include ligand potency ($IC_{50}$ value) and ligand cobinding cooperativity (hill slope). Ligand cobinding potencies were also compared to the non-covalent binding affinities measured using a TR-FRET ligand displacement assay.

The MRL-24 assays (*Figure 5a*) show that cobinding potency decreases with a general rank ordering of covalent inhibitor efficacy GW9662 < T0070907 < SR33065 < SR36708. Relative to MRL-24-binding affinity to apo-PPARγ LBD, the cobinding potencies are right-shifted, or weaker, by two to six orders of magnitude. Increasing NCoR1 peptide concentrations in the assays further enhances weakening of MRL-24 cobinding potency. MRL-24 cobinding is non-cooperative (hill slopes near –1) for most conditions and not significantly affected by the different NCoR1 peptide concentrations. These data

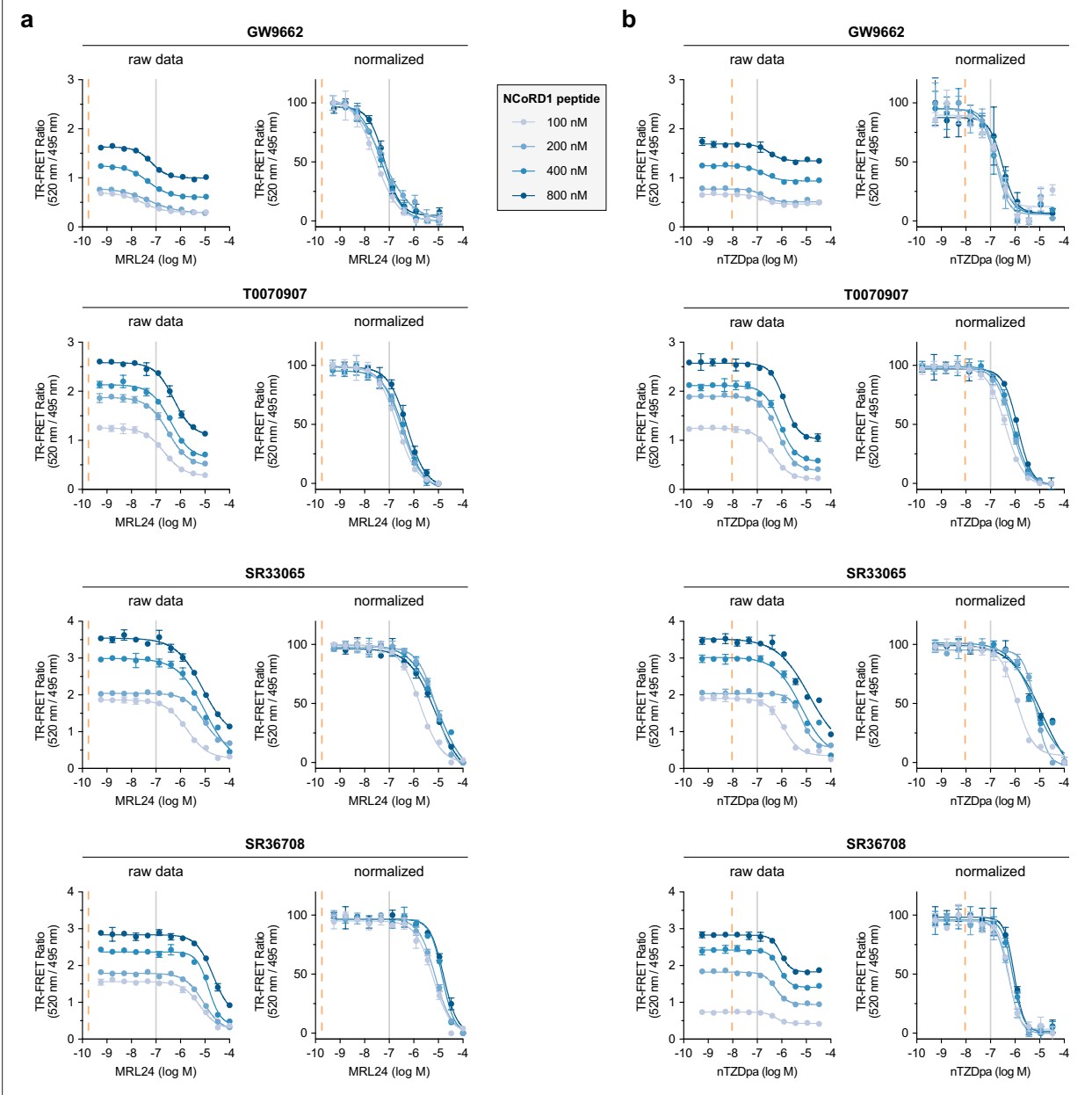

**Figure 4.** Time-resolved fluorescence resonance energy transfer (TR-FRET) assay profiling of non-covalent ligand cobinding to PPARγ ligand-binding domain (LBD) with a covalent inhibitor and increasing NCoR1 peptide concentrations. TR-FRET NCoR1 corepressor peptide interaction assays where (**a**) MRL-24 or (**b**) nTZDpa were titrated with increasing concentrations of FITC-labeled NCoR1 corepressor peptide to saturate the peptide-bound forms of PPARγ LBD bound to the covalent ligands profiled in *Figure 1* (*n* = 3 technical replicates; mean ± SD). MRL-24 and nTZDpa $K_i$ values are noted with vertical dotted orange lines, and a vertical gray line denotes log *M* = −7 as a visual cue to compare the concentration–response curves.

indicate that covalent ligands with improved pharmacological inverse agonist efficacy, in particular SR36708, are more effective inhibitors of MRL-24 cobinding.

For the nTZDpa assays (*Figure 5b*), cobinding potency values in the presence of the covalent inhibitors follow similar trends as the MRL-24 assays—except for SR36708, which is nearly as effective as T0070907 but not as effective as SR33065 for inhibiting nTZDpa cobinding. Relative to nTZDpa-binding affinity to apo-PPARγ LBD, the cobinding potencies are also right-shifted but only to a lesser degree than MRL-24, approximately one to two orders of magnitude. Similar to MRL-24, increasing NCoR1 peptide concentrations in the assays further enhances weakening of nTZDpa cobinding

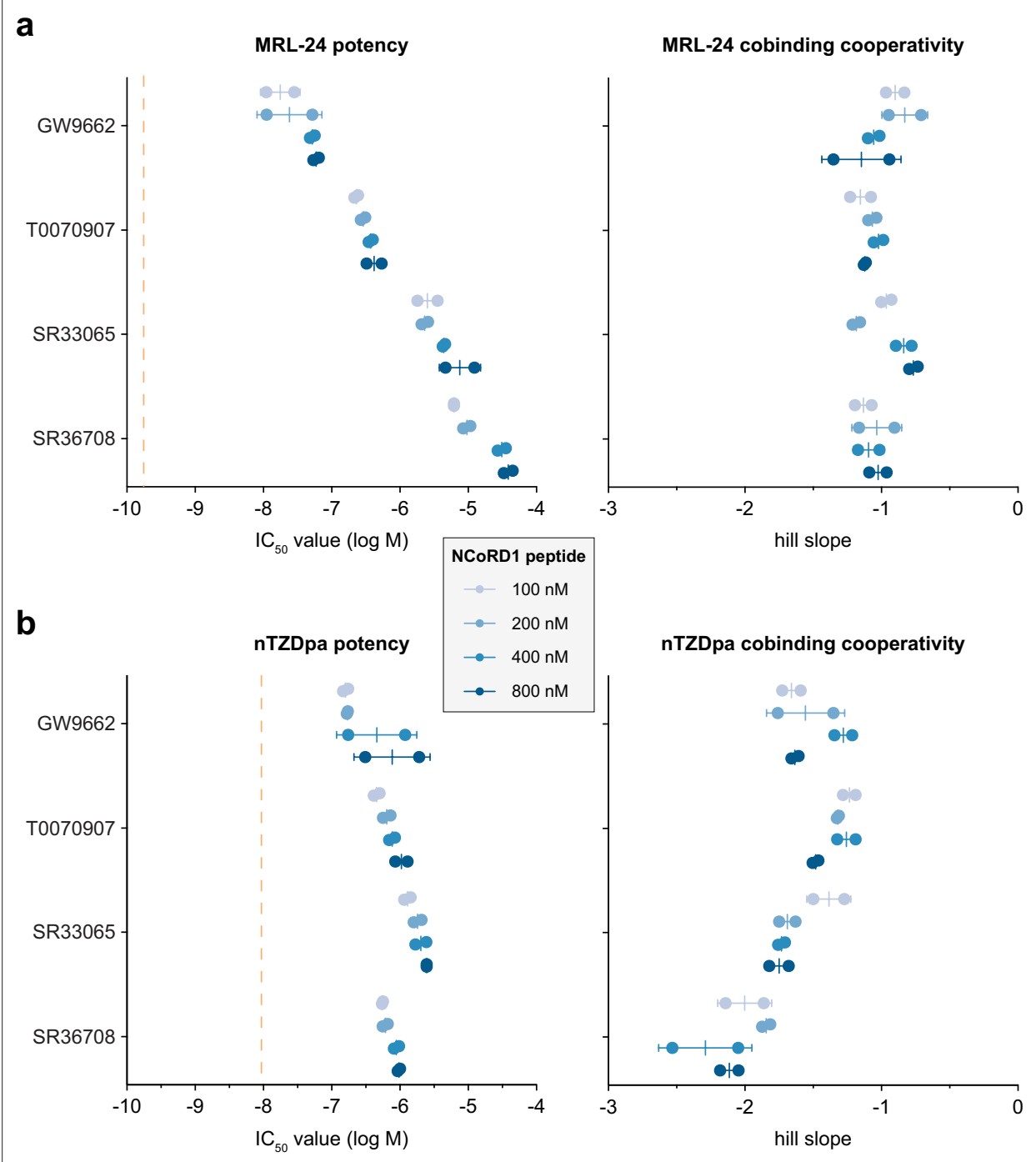

**Figure 5.** Corepressor peptide binding synergizes with covalent inhibitor inverse agonism to weaken non-covalent ligand cobinding. Fitted parameters extracted from time-resolved fluorescence resonance energy transfer (TR-FRET) NCoR1 ligand cobinding assays (*Figure 5—source data 1*) with (**a**) MRL-24 or (**b**) nTZDpa including potency ($IC_{50}$) and cooperativity (hill slope) values ($n = 2$ biological replicates derived from a fit of TR-FRET data with $n = 3$ technical replicates; mean ± SD). MRL-24 and nTZDpa Ki values are noted with vertical dotted orange lines.

The online version of this article includes the following source data for figure 5:

**Source data 1.** Data underlying the plots in *Figure 5*.

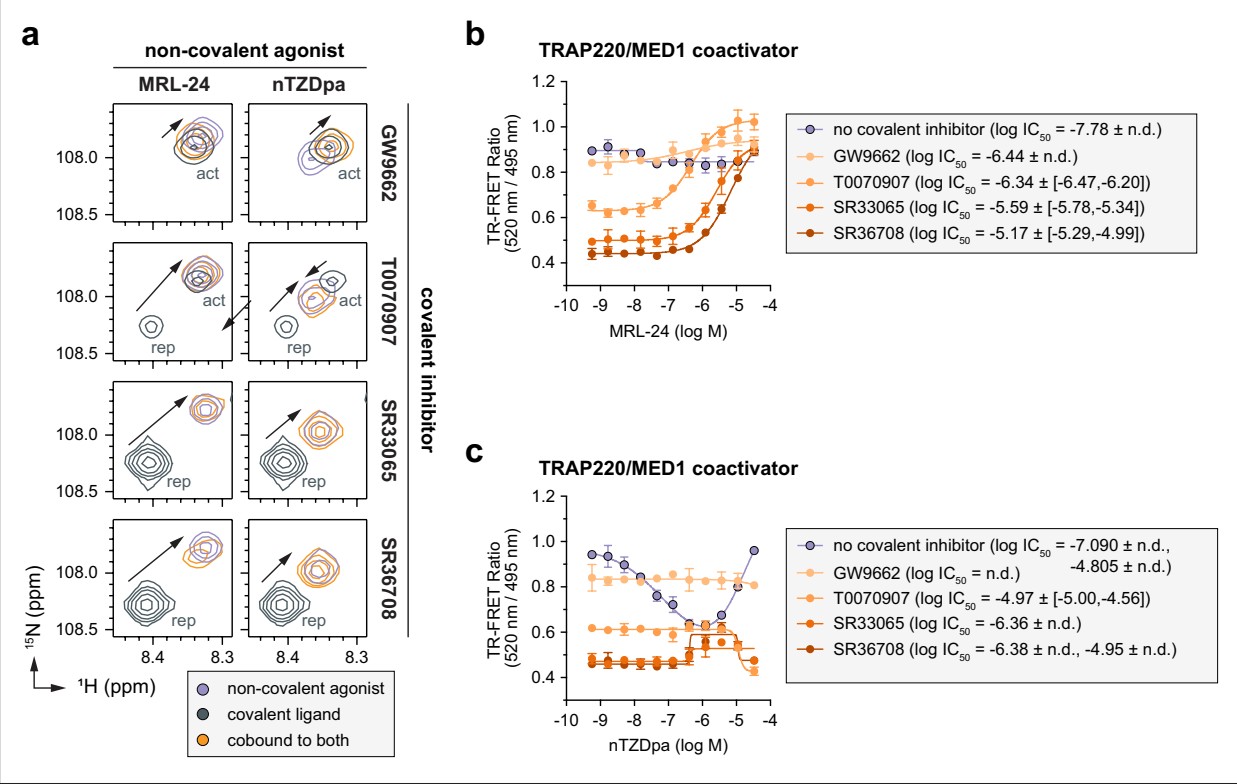

**Figure 6.** Non-covalent ligand and covalent inhibitor cobinding shifts the PPARγ ligand-binding domain (LBD) conformational ensemble to an active state. (**a**) 2D [¹H,¹⁵N]-TROSY-HSQC data zoomed into the Gly399 backbone amide peaks of ¹⁵N-labeled PPARγ LBD bound individually or cobound to the indicated non-covalent and covalent ligands. The active (act) and repressive (rep) peaks when bound to covalent ligand alone are labeled; black arrows denote the shift in the peaks between the covalent ligand bound alone vs. cobound with the non-covalent ligand. Time-resolved fluorescence resonance energy transfer (TR-FRET) TRAP220/MED1 coactivator peptide interaction assays where (**b**) MRL-24 or (**c**) nTZDpa were titrated into PPARγ LBD pretreated with the indicated covalent inhibitors (*n* = 3 technical replicates; mean ± [95% CI]; n.d. = not determined).

The online version of this article includes the following figure supplement(s) for figure 6:

**Figure supplement 1.** 2D [¹H,¹⁵N]-TROSY-HSQC NMR data of ¹⁵N-labeled PPARγ ligand-binding domain (LBD) bound individually or cobound to MRL-24 (top), nTZDpa (bottom), and GW9662 as indicated.

**Figure supplement 2.** 2D [¹H,¹⁵N]-TROSY-HSQC NMR data of ¹⁵N-labeled PPARγ ligand-binding domain (LBD) bound individually or cobound to MRL-24 (top), nTZDpa (bottom), and T0070907 as indicated.

**Figure supplement 3.** 2D [¹H,¹⁵N]-TROSY-HSQC NMR data of ¹⁵N-labeled PPARγ ligand-binding domain (LBD) bound individually or cobound to MRL-24 (top), nTZDpa (bottom), and SR33065 as indicated.

**Figure supplement 4.** 2D [¹H,¹⁵N]-TROSY-HSQC NMR data of ¹⁵N-labeled PPARγ ligand-binding domain (LBD) bound individually or cobound to MRL-24 (top), nTZDpa (bottom), and SR36708 as indicated.

potency. However, nTZDpa cobinding occurs with positive cooperativity, as the fitted hill slopes are more negative than –1, in particular for SR36708 with hill slopes near or greater than –2.

## Ligand cobinding with covalent inverse agonists stabilizes an active LBD conformation

To structurally validate the ligand cobinding we observed in the TR-FRET assays, we compared 2D [¹H,¹⁵N]-TROSY-HSQC NMR data of ¹⁵N-labeled PPARγ LBD bound to the covalent inhibitor or non-covalent agonist alone or cobound to both (*Figure 6a*, *Figure 6—figure supplements 1–4*). Focusing on Gly399 as a proxy for the AF-2 surface conformation, MRL-24 or nTZDpa cobinding to LBD preincubated with GW9662 shows CSPs that shift to lower ¹H and ¹⁵N chemical shift values. Ligand cobinding to LBD preincubated with T0070907 shows a similar CSP shift for MRL-24, but for nTZDpa, the peak is shifted in between the repressive and active T0070907-bound states. We previously showed that this upward and right-shifting CSP pattern, where NMR peaks move along a

diagonal between transcriptional active and repressive states (*MacTavish et al., 2025*), corresponds to stabilizing an active LBD conformation that we visualized in crystal structures of the PPARγ LBD (*Shang et al., 2019*; *Shang and Kojetin, 2024*). For T0070907-bound LBD, which exchanges between two long-lived active and repressive states in slow exchange on the NMR time scale resulting in the appearance of two NMR peaks for Gly399, ligand cobinding resulting in the disappearance of the repressive state, a shift in the LBD conformation in an active state. The shift in LBD conformation was particularly noticeable for T0070907-bound LBD compared to GW9662-bound LBD, which only populates the active-like state on its own (*Brust et al., 2018*). Ligand cobinding to $^{15}$N-labeled PPARγ LBD preincubated with SR33065 or SR36708, which on their own fully stabilize the LBD in a repressive conformation, also shifts the LBD toward an active state. In most cases, the ligand cobound active state NMR peak overlaps with the NMR peak corresponding bound to MRL-24 or nTZDpa alone, which indicates the active PPARγ LBD conformation is similar whether the non-covalent ligand is bound alone or cobound with a covalent inhibitor.

We previously showed that non-covalent ligand cobinding with GW9662 or T0070907 can activate PPARγ-mediated transcription and increase the expression of PPARγ target genes (*Hughes et al., 2014*)—a cellular activation phenotype that correlates to increased recruitment of TRAP220/MED1 coactivator peptide in biochemical TR-FRET coregulator interaction assays (*Hughes et al., 2014*; *MacTavish et al., 2025*). Because the NMR data indicate that ligand cobinding stabilizes an active LBD conformation, we performed TR-FRET assays using an FITC-labeled TRAP220/MED1 coactivator peptide to determine the potential for ligand cobinding to promote agonist-like behavior. In the absence of a covalent inhibitor, titration of MRL-24 resulted in no significant change in the TRAP220/MED1 TR-FRET coregulator recruitment assay (*Figure 6b*). nTZDpa showed biphasic behavior, decreasing TRAP220/MED1 interaction at high concentrations and increasing TRAP220/MED1 at lower concentrations (*Figure 6c*). However, in the presence of a covalent inhibitor, MRL-24 showed a concentration-dependent increase in TRAP220/MED1 binding with TR-FRET ratio efficacy values plateauing at values similar to, or higher than, MRL-24 titrated without a covalent inhibitor. In contrast, nTZDpa showed a small increase in TRAP220/MED1 binding in the presence of SR33065 or SR36708, but not GW9662 or T0070907; and a small decrease in TRAP220/MED1 binding at higher nTZDpa concentrations—a profile opposite of nTZDpa titrated on its own. These TRAP220/MED1 TR-FRET data are consistent with the NMR data where MRL-24 cobinding with a covalent inhibitor shifts the LBD toward an active conformation to a larger degree than nTZDpa cobinding.

## Ligand cobinding is not blocked by SR16832, a covalent inhibitor that functions via a different mechanism

The data above indicate that covalent inhibitors with improved pharmacological inverse agonism functions can further weaken ligand cobinding compared to GW9662 and T0070907, but none of these inhibitors completely block ligand binding to the PPARγ LBD. We next sought to determine if MRL-24 and nTZDpa binding can be inhibited by a commercially available PPARγ dual-site covalent inhibitor called SR16832, which inhibited binding of some PPARγ agonists including rosiglitazone and a chemical analog of MRL-24 called MRL-20 in biochemical and cellular assays (*Brust et al., 2017*). FP coregulator peptide assays reported that SR16832 does not significantly affect NCoR1 or TRAP220/MED1 peptide-binding affinity compared to DMSO control. However, SR16832 represses PPARγ-mediated transcription on its own to levels approaching or similar to T0070907-treated cells. Taken together with the FP data, these cellular data suggest that SR16832 may function by inhibiting ligand binding via direct clashing as opposed to the allosteric mechanism linked to pharmacological covalent inverse agonism.

We compared 2D [$^{1}$H,$^{15}$N]-TROSY-HSQC NMR data of $^{15}$N-labeled PPARγ LBD bound to SR16832 inhibitor, non-covalent agonist alone, or cobound to both (*Figure 7—figure supplement 1*), which revealed that MRL-24 and nTZDpa can still bind in the presence of SR16832. Focusing on Gly399 again as a proxy for the structural conformation of the AF-2 surface (*Figure 7a*), nTZDpa cobinding shows a small CSP that moves off the active-to-repressive NMR CSP diagonal (*MacTavish et al., 2025*) relative to MRL-24 cobinding, which shows a larger CSP along the diagonal to higher $^{1}$H and $^{15}$N chemical shift values indicating a shift in LBD conformation away from an active state and toward a repressive state. TR-FRET assays using PPARγ LBD pretreated with SR16832 and the FITC-labeled NCoR1 corepressor peptide at two concentrations revealed no detectable binding for MRL-24 but

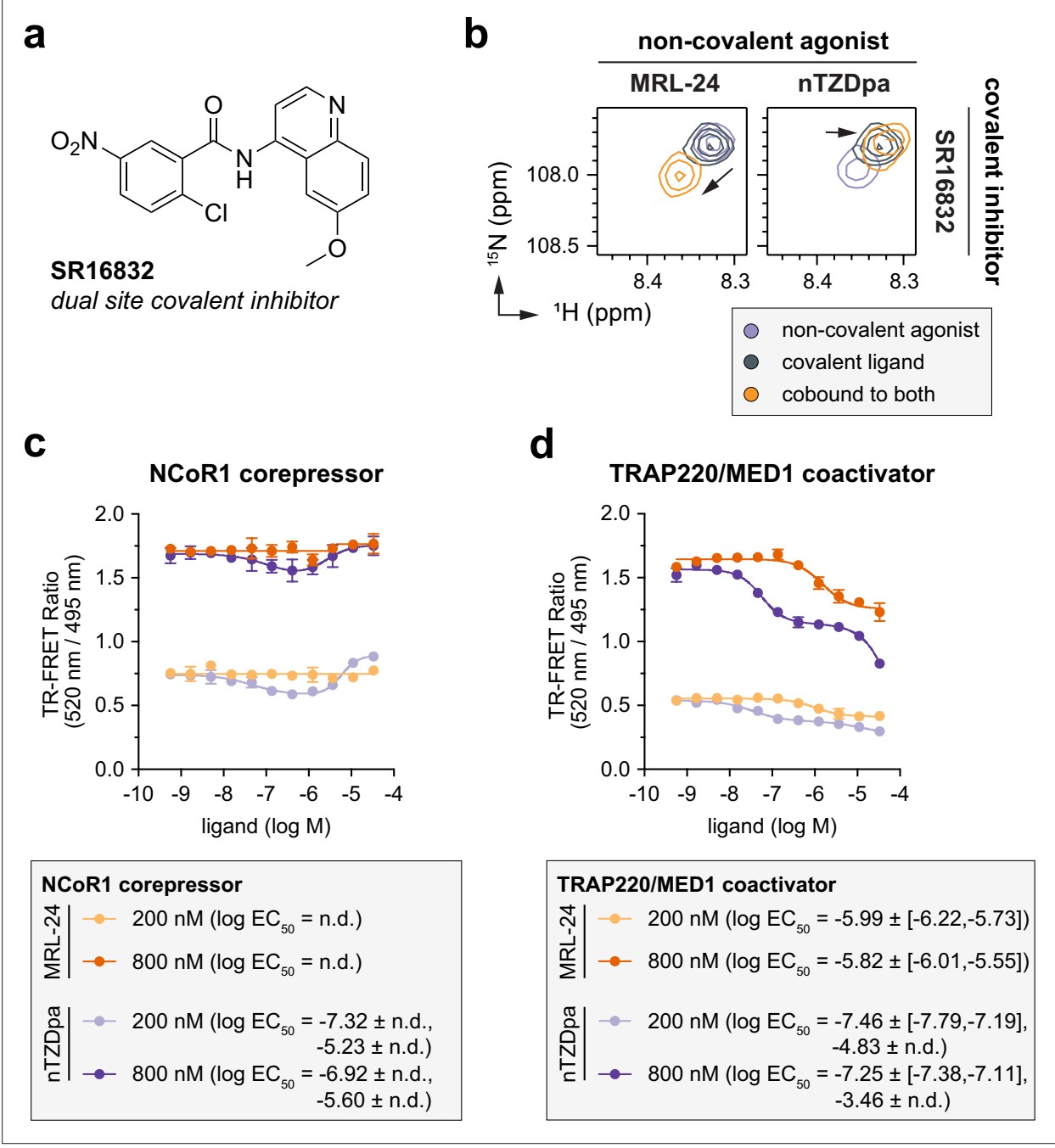

**Figure 7.** Non-covalent ligand cobinding occurs in the presence of the covalent inhibitor SR16832. (**a**) Chemical structure of the dual-site covalent inhibitor SR16832. (**b**) 2D [$^1$H,$^{15}$N]-TROSY-HSQC data zoomed into the Gly399 backbone amide peaks of $^{15}$N-labeled PPARγ ligand-binding domain (LBD) bound individually or cobound to MRL-24, nTZDpa, and SR16832 as indicated. Black arrows denote the shift in the peaks, or LBD conformation, between the covalent ligand bound alone vs. cobound with the non-covalent ligand. Time-resolved fluorescence resonance energy transfer (TR-FRET) coregulator interaction assays using (**c**) FITC-labeled NCoR1 corepressor or (**d**) FITC-labeled TRAP220/MED1 coactivator peptides where MRL-24 (orange) or nTZDpa (purple) were titrated with two concentrations of FITC-NCoR1 corepressor peptide to saturate the peptide-bound forms of PPARγ LBD bound to SR16832 (*n* = 3 technical replicates; mean ± SD).

The online version of this article includes the following figure supplement(s) for figure 7:

**Figure supplement 1.** 2D [$^1$H,$^{15}$N]-TROSY-HSQC NMR data of $^{15}$N-labeled PPARγ ligand-binding domain (LBD) bound individually or cobound to MRL-24 (top), nTZDpa (bottom), and SR16832 as indicated.

a concentration-dependent biphasic binding effect for nTZDpa (*Figure 7b*). Furthermore, TR-FRET assays using an FITC-labeled TRAP220/MED1 coactivator peptide showed binding effects for both MRL-24 and nTZDpa (*Figure 7c*). For MRL-24, ligand cobinding with SR16832 resulted in displacement of both NCoR1 and TRAP220/MED1 peptides, which is consistent with an antagonist-like pharmacological function and the NMR data showing that MRL-24 cobinding with SR16832 shifts the LBD ensemble away from an active state. The relationship between TR-FRET and NMR data for nTZDpa is more nuanced. The TR-FRET data indicate two equivalents of nTZDpa may bind to SR16832-bound PPARγ LBD where both binding events inhibit TRAP220/MED1 interaction, but NCoR1 interaction is decreased by the first binding event and increased by the second binding event. Taken together, these data indicate that it is possible for ligand cobinding to occur when PPARγ LBD is pretreated with a covalent inhibitor, in some cases without causing a significant impact on coregulator peptide affinity that would result in a change in TR-FRET signal (e.g., MRL-24 and NCoR1 peptide).

## Discussion

In the nuclear receptor field, small molecule antagonists that bind at the same site as the orthosteric endogenous ligand are used, when available, as pharmacological chemical tools to probe the functions of blocking ligand-induced nuclear receptor activities. However, aside from their use as inhibitors of ligand binding, antagonists can also display pharmacological properties on their own. Pharmacological nuclear receptor ligands are classified based on the molecular basis by which they influence transcription. Agonists recruit coactivator proteins to activate transcription, inverse agonists recruit corepressor proteins to repress transcription, and the mechanisms by which antagonists function are not fully understood but are thought to involve competition with an endogenous ligand resulting in no change in coregulator interaction and transcription or a decrease in transcription without recruiting a corepressor protein (*Strutzenberg et al., 2019*). GW9662 and T0070907 were first reported as antagonist inhibitors of ligand binding to PPARγ (*Lee et al., 2002*; *Leesnitzer et al., 2002*). However, more than a decade later, we showed these antagonist inhibitors do not in fact block binding of all PPARγ

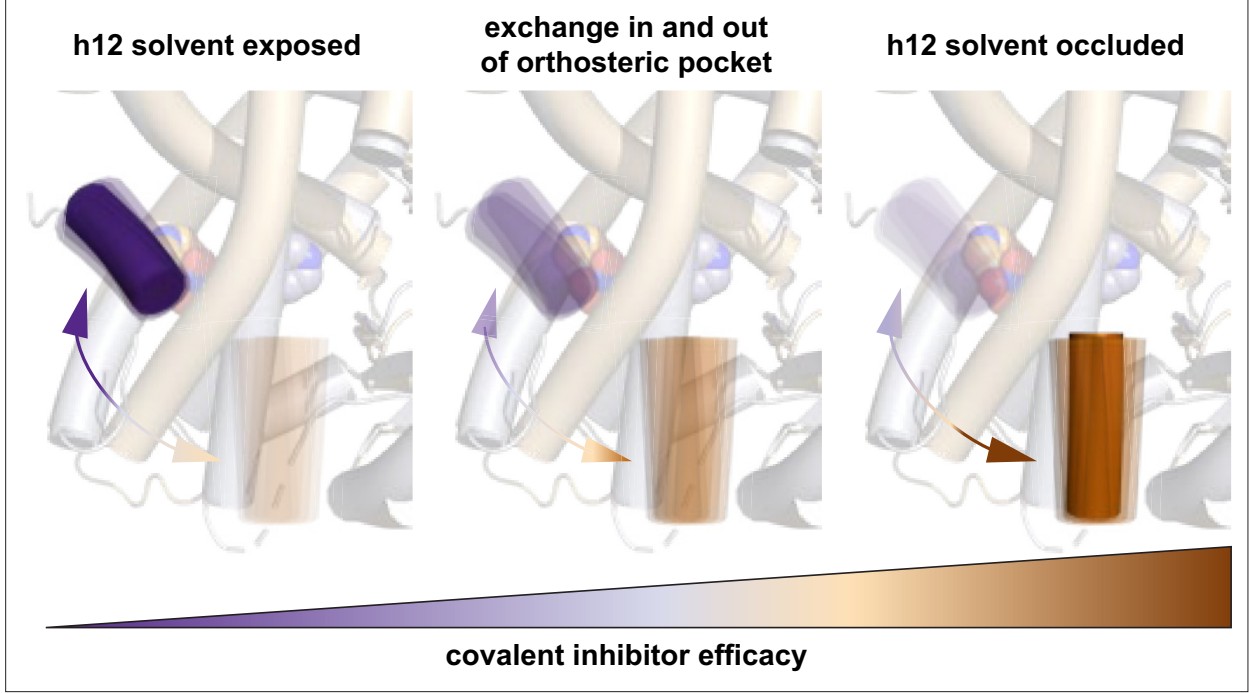

**Figure 8.** Model for improved covalent inhibitor efficacy. Covalent ligands that when bound to PPARγ ligand-binding domain (LBD) increase the occupancy of helix 12 (h12) within the orthosteric pocket show improved efficacy of inhibiting other ligands from binding to the orthosteric pocket. Our data suggest that this is an allosteric mechanism in the sense that the covalent inhibitor does not physically clash with the non-covalent ligand itself; instead, increased occupancy of h12 within the orthosteric pocket resulting from the bound covalent inhibitor results in a clash or competition with a non-covalent orthosteric ligand.

ligands (*Hughes et al., 2014*) and display unique pharmacological activities (*Brust et al., 2018*) linked to their ability to stabilize the PPARγ LBD in a transcriptionally repressive conformation with helix 12 docked in the orthosteric pocket (*Shang et al., 2020*).

The PPARγ LBD is a dynamic conformational ensemble that exchanges between transcriptionally active and repressive conformations in the absence of ligand (*Shang et al., 2020*). Ligand binding can shift the LBD conformational ensemble between graded agonist and graded inverse agonist states (*MacTavish et al., 2025*; *Shang et al., 2019*). We recently proposed an allosteric mechanism that describes how T0070907 is a more effective inhibitor of weakening orthosteric ligand binding than GW9662, not by direct clashing of these covalent inhibitors with an orthosteric ligand but via increased occupancy of helix 12 within the orthosteric pocket (*Shang and Kojetin, 2024*). Because T0070907 partially stabilizes the LBD in a repressive state that can exchange back to the active state, we hypothesized that covalent compounds such as SR33065 and SR36708 that more fully stabilize the repressive LBD state where helix 12 occupies the orthosteric pocket (*MacTavish et al., 2025*) may completely block ligand binding to the orthosteric pocket (*Shang and Kojetin, 2024*; *Figure 8*). Although our data here support this hypothesis, our data here show that SR33065 and SR36708 weaken but do not completely inhibit ligand binding. This observation indicates that the allosteric mechanism of stabilizing the LBD in a repressive conformation with helix 12 located in the orthosteric pocket is not enough to prevent a ligand from binding and, in doing so, likely pushes helix 12 out of the repressive conformation into an active conformation—similar to the two-step binding mechanism we reported for agonist binding (*Shang and Kojetin, 2021*).

The allosteric ligand inhibition mechanism afforded by the pharmacological ligand series of GW9662, T0070907, SR33065, and SR36708 appears to differ from a different covalent PPARγ ligand, SR16832, which by NMR does not stabilize a repressive helix 12 conformation and would therefore likely inhibit binding due to direct clashing with other orthosteric ligands (*Brust et al., 2017*). However, our NMR data show that MRL-24 and nTZDpa can still bind to the PPARγ LBD that is pretreated with the SR16832 as a covalent inhibitor. Notably, our TR-FRET data show that nTZDpa cobinding is weakened when PPARγ LBD is pretreated with SR16832; however, MRL-24 shows no change in TR-FRET signal even though the NMR data shows that MRL-24 cobinds to SR16832-bound PPARγ LBD. This finding highlights an important warning for the field: functional assays that are used as a proxy for ligand binding, including TR-FRET coregulator interaction biochemical assays or luciferase reporter assays, may not be well suited to detect actual and direct ligand binding such as methods like protein NMR. This is of importance because in the PPARγ field, it is well known that a high-affinity ligand can bind and produce no change in luciferase reporter activity yet still affect gene expression (*Choi et al., 2011*). It is possible that by using peptides derived from different coactivators or corepressors, or different segments of the same coregulator—as certain segments of the same coregulator protein can bind to an NR LBD with high affinity and others bind with low affinity or not at all—subtle changes caused by ligand cobinding may be detected. Furthermore, biochemical assays using fluorescent or radiolabeled tracer ligands as a proxy for measuring ligand-binding affinity may also miss ligand cobinding events—once the tracer ligand has been displaced, or blocked by an antagonist, the assay is incapable of detecting any additional ligand-binding events such as ligand cobinding.

It is not yet clear if other nuclear receptors share the same ligand-induced activity-dependent conformational ensemble to postulate if our findings here on PPARγ can translate to other nuclear receptors. However, our findings are likely applicable to other non-covalent orthosteric PPARγ ligands regardless of agonist, antagonist, or inverse agonist pharmacology—and generally suggest covalent ligand that stabilizes helix 12 within orthosteric pocket may function in some degree to inhibit binding of other orthosteric ligands.

Our observations also raise the question as to how to design a completely effective inhibitor of ligand binding to PPARγ. It is possible that covalent analogs of SR33065 and SR36708 that not only stabilize a repressive conformation but also contain an additional covalent warhead to lock the repressive helix 12 in place may provide a route to design improved covalent inhibitors. It is also possible that SR16832 could be used as a parent compound to discover a covalent inhibitor that likely occupies more of the orthosteric pocket and the so-called alternate site region corresponding to the entrance cavity to the orthosteric pocket (*Hughes et al., 2016*; *Hughes et al., 2014*; *Shang et al., 2018*). Thus, although our data here provide further support to the allosteric ligand inhibition model, our findings also reveal a significant unmet need in the field and a word of caution. Three of the covalent ligands

described here are commercially available and used by the field (GW9662, T0070907, and SR16832), but are not effective inhibitors of ligand binding to PPARγ—and they have pharmacological functions distinct from the inability to effectively block ligand binding.

## Materials and methods

**Key resources table**

| Reagent type (species) or resource | Designation | Source or reference | Identifiers | Additional information |
|---|---|---|---|---|
| Gene (*Homo sapiens*) | PPARG | UniProt | PPARG_HUMAN | Residues 203–477, isoform 1 numbering |
| Strain, strain background (*Escherichia coli*) | BL21(DE3) | New England Bioscience | #C2527H | Electrocompetent cells used for bacterial expression of proteins |
| Cell line (*Homo sapiens*) | HEK293T | ATCC | #CRL-3216; RRID:CVCL_0063 | Used for luciferase reporter assays |
| Antibody | LanthaScreen Elite Tb-anti-His Antibody | Thermo Fisher Scientific | #PV5895; RRID:AB_3720338 | TR-FRET antibody |
| Recombinant DNA reagent | Human PPARγ LBD - pET46 (plasmid) | *Hughes et al., 2012* | | Used to express human PPARγ ligand-binding domain (LBD) protein |
| Peptide, recombinant protein | Human NCoR1 ID2 motif | LifeTein | | FITC-labeled peptide for TR-FRET |
| Peptide, recombinant protein | Human TRAP220/MED1 ID2 motif | LifeTein | | FITC-labeled peptide for TR-FRET |
| Chemical compound, drug | T0070907 | Cayman Chemical | #10026 | Covalent PPARγ inverse agonist |
| Chemical compound, drug | GW9662 | Cayman Chemical | #70785 | Covalent PPARγ antagonist |
| Chemical compound, drug | SR33065 | *MacTavish et al., 2025* | | Covalent PPARγ inverse agonist |
| Chemical compound, drug | SR33068 | *MacTavish et al., 2025* | | Covalent PPARγ inverse agonist |
| Chemical compound, drug | MRL-24 | Abbexa | #abx282275 | PPARγ partial agonist |
| Chemical compound, drug | nTZDpa | Tocris | #2150 | PPARγ partial agonist |
| Chemical compound, drug | Rosiglitazone | Cayman Chemical | #71740 | PPARγ agonist |
| Software, algorithm | Prism | GraphPad | Version 10 | Used to plot assay data |
| Software, algorithm | PyMOL | Schrödinger | Version 3 | Used to generate structural plots |
| Other | X-tremegene 9 | Roche | | Transfection reagent |
| Other | Britelite Plus | PerkinElmer | | Luciferase reagent |

### Materials and reagents

All compounds used in this study were obtained from commercial sources including Cayman Chemicals, Tocris Bioscience, and Sigma-Aldrich with purity levels >95%; or were previously reported and validated (*MacTavish et al., 2025*) for identity and purity levels >95%. Peptides of NCoR1 ID2 motif (2256–2278; DPASNLGLEDIIRKALMGSFDDK) or human TRAP220/MED1 ID2 motif (residues 638–656; NTKNHPMLM NLLKDNPAQD) with an N-terminal FITC label with a six-carbon linker (Ahx) and an amidated C-terminus for stability were synthesized by LifeTein. Ligands including GW9662, MRL-24, nTZDpa, rosiglitazone, and T0070907 were obtained from commercial vendors including Abbexa, Cayman Chemical, and Tocris; or, in the case of SR33065 and SR36708, were previously reported and characterized by our labs (*MacTavish et al., 2025*).

### Protein expression and purification

Human PPARγ LBD (residues 203–477, isoform 1 numbering) was expressed in *Escherichia coli* BL21(DE3) cells using autoinduction ZY media as a Tobacco Etch Virus (TEV)-cleavable N-terminal His-tagged fusion protein using a pET46 Ek/LIC vector (Novagen) and purified using Ni-NTA affinity chromatography and gel filtration chromatography. The purified proteins were typically concentrated to

10 mg/ml in a buffer consisting of 20 mM potassium phosphate (pH 7.4), 50 mM potassium chloride, 5 mM tris(2-carboxyethyl)phosphine (TCEP), and 0.5 mM ethylenediaminetetraacetic acid (EDTA). Purified protein was verified by SDS–PAGE as >95% pure. For studies using a covalent ligand, typically PPARγ LBD protein was incubated with at least a ~1.05x excess of covalent ligand at 4°C for 24 hr to ensure covalent modification to residue Cys285 (complete attachment of the covalent ligands typically occurs within 30–60 min) then buffer exchanged the sample to remove excess covalent ligand and DMSO.

## Cellular luciferase reporter assays

Using methods we previously reported (*MacTavish et al., 2025*), HEK293T cells obtained from ATCC (#CRL-3216) and free of mycoplasma contamination were cultured in Dulbecco's minimal essential medium (DMEM, Gibco) supplemented with 10% fetal bovine serum and 50 units ml$^{-1}$ of penicillin, streptomycin, and glutamine. Cells were grown to 90% confluency in T-75 flasks; from this, 2 million cells were seeded in a 10 cm cell culture dish for transfection using X-tremegene 9 (Roche) and Opti-MEM (Gibco) with full-length human PPARγ isoform 2 expression plasmid (4 µg), and a luciferase reporter plasmid containing the three copies of the PPAR-binding DNA response element (PPRE) sequence (3xPPRE-luciferase; 4 µg). After an 18-hr incubation, cells were transferred to white 384-well cell culture plates (Thermo Fisher Scientific) at 10,000 cells/well in 20 µl total volume/well. After a 4-hr incubation, cells were treated in quadruplicate with 20 µl of either vehicle control (1.5% DMSO in DMEM media) or 5 µM ligand, or in dose–response format to determine EC50/IC50 values, where each ligand treated condition had separate control wells to account for plate location-based artifacts. After a final 18-hr incubation, cells were harvested with 20 µl Britelite Plus (PerkinElmer), and luminescence was measured on a BioTek Synergy Neo multimode plate reader. Data were plotted in GraphPad Prism as luciferase activity vs. ligand concentration (*n* = 4 technical replicates). Data shown are representative of two or more independent experiments (biological replicates).

## FP coregulator interaction assays

Using methods we previously reported (*MacTavish et al., 2025*), FP assays were performed using 6xHis-PPARγ LBD preincubated with or without 2 molar equivalents of compound at 4°C overnight and buffer exchanged via centrifugal concentration using Amicon Ultra centrifugal filters to remove excess ligand. Protein samples were serially diluted into a buffer containing 20 mM potassium phosphate (pH 8), 50 mM potassium chloride, 5 mM TCEP, 0.5 mM EDTA, and 0.01% Tween-20 and plated with 180 nM FITC-labeled NCoR1 ID2 or FITC-labeled TRAP220/MED1 ID2 peptide in black 384-well plates (Greiner). The plate was incubated at 25°C for 1 hr, and FP was measured on a BioTek Synergy Neo multimode plate reader at 485 nm emission and 528 nm excitation wavelengths. Data were plotted using GraphPad Prism as FP signal in millipolarization units vs. protein concentration (n=3 experimental replicates); fit to a one site—total binding equation using a consistent, fixed Bmax value determined from a fit of the high affinity interactions as binding for some conditions did not saturate at the highest protein concentration used (45 µM). Data shown are representative of two or more independent experiments (biological replicates).

## TR-FRET assays

Using methods similar to those we previously reported (*MacTavish et al., 2025*; *Shang et al., 2019*), TR-FRET assays were performed in black 384-well plates (Greiner) with 23 µl final well volume. For coregulator interaction assays, each well contained 4 nM 6xHis-PPARγ LBD with or without covalent ligand modification, 1 nM LanthaScreen Elite Tb-anti-His Antibody (Thermo Fisher), and NCoR1 peptide as indicated (100, 200, 400, or 800 nM) or TRAP220/MED1 peptide (400 nM) in a buffer containing 20 mM KPO$_4$ pH 7.4, 50 mM KCl, 5 mM TCEP, 0.005% Tween 20. For the ligand displacement assays, each well contained 1 nM 6xHis-PPARγ LBD protein, 1 nM LanthaScreen Elite Tb-anti-HIS Antibody (Thermo Fisher Scientific), and 5 nM Fluormone Pan-PPAR Green (Invitrogen) in a buffer containing 20 mM KPO$_4$ (pH 8), 50 mM KCl, 5 mM TCEP, and 0.005% Tween-20. Compound stocks were prepared via serial dilution in DMSO, added to wells in triplicate, and plates were read using BioTek Synergy Neo multimode plate reader after incubating at 25°C for 1 hr. The Tb donor was excited at 340 nm, its emission was measured at 495 nm, and the acceptor FITC emission was measured at 520 nm. Data were plotted and fit to equations using GraphPad Prism as TR-FRET ratio

520/495 nm vs. ligand concentration ($n$ = 3 experimental replicates). Coregulator interaction data were fit to a four-parameter sigmoidal dose–response equation—or biphasic or bell-shaped dose–response equations when appropriate—determined by comparison of fits to both equations and $F$ test where the simpler model is selected if the p value is less than 0.05. Ligand displacement data were fit to the one site—Fit $K_i$ binding equation to obtain $K_i$ values using the reported binding affinity of Fluormone Pan-PPAR Green (2.8 nM; Invitrogen PV4894 product insert). Data shown are representative of two or more independent experiments (biological replicates).

## NMR spectroscopy

Two-dimensional [$^1$H,$^{15}$N]-TROSY HSQC NMR data of $^{15}$N-labeled PPARγ LBD (200 µM) were acquired at 298 K on a Bruker 700 or 900 MHz NMR instrument equipped with a QCI or TCI cryoprobe, respectively, in NMR buffer (50 mM potassium phosphate, 20 mM potassium chloride, 1 mM TCEP, pH 7.4, 10% $D_2O$). Covalent ligands were preincubated overnight at 4°C with 2 molar equivalents and buffer exchanged to remove excess ligand. Non-covalent ligands were either added at 1 molar equivalent alone or 2 molar equivalents when cobound to a covalent ligand. Data were collected using Topspin 3.0 (Bruker Biospin) and processed/analyzed using NMRFx (*Norris et al., 2016*) or Bruker Topspin. NMR chemical shift assignments previously transferred from rosiglitazone-bound PPARγ LBD (*Hughes et al., 2012*) to T0070907- and GW9662-bound states (*Brust et al., 2018*; *Shang et al., 2020*) were used in this study for well-resolved residues with conserved NMR peak positions to the previous ligand-bound forms using the minimum CSP procedure (*Williamson, 2013*).

## Acknowledgements

This work was supported in part by the National Institutes of Health (NIH) grant R01DK124870 from the National Institute of Diabetes and Digestive and Kidney Diseases (NIDDK). NMR instrumentation at Vanderbilt is supported in part by grants from the NSF-MRI (0922862), acquisition of a 900 MHz Ultra-High Field NMR spectrometer in 2009; NIH (S10RR025677) for console upgrades on all biomolecular NMR spectrometers in 2009; NIH (R35GM118089-04S1), a Vanderbilt Trans-Institutional Programs (TIPs) grant to purchase the IVDr equipment in 2019; NIH supplement for the helium liquefier in 2019; NIH (S10OD034276) to replace the 800 MHz spectrometer in 2024 accompanied by matching funds from Vanderbilt University. The contents of this publication are solely the responsibility of the authors and do not necessarily represent the official views of NIDDK, NIH, or NSF.

## Additional information

### Funding

| Funder | Grant reference number | Author |
|---|---|---|
| National Institute of Diabetes and Digestive and Kidney Diseases | R01DK124870 | Douglas J Kojetin |

The funders had no role in study design, data collection, and interpretation, or the decision to submit the work for publication.

### Author contributions

Liudmyla Arifova, Brian S MacTavish, Jinsai Shang, Min-Hsuan Li, Xiaoyu Yu, Formal analysis, Investigation, Writing – review and editing; Zane Laughlin, Mithun Nag Karadi Girdhar, Di Zhu, Investigation, Writing – review and editing; Theodore M Kamenecka, Supervision, Validation, Methodology, Project administration, Writing – review and editing; Douglas J Kojetin, Conceptualization, Formal analysis, Supervision, Funding acquisition, Validation, Visualization, Methodology, Writing – original draft, Project administration, Writing – review and editing

### Author ORCIDs

Liudmyla Arifova https://orcid.org/0009-0009-3268-8923
Brian S MacTavish https://orcid.org/0000-0002-6341-6244

Mithun Nag Karadi Girdhar ⓘ https://orcid.org/0000-0003-0520-5921
Jinsai Shang ⓘ https://orcid.org/0000-0001-8164-1544
Xiaoyu Yu ⓘ https://orcid.org/0000-0003-0549-9560
Theodore M Kamenecka ⓘ https://orcid.org/0000-0002-3077-0167
Douglas J Kojetin ⓘ https://orcid.org/0000-0001-8058-6168

Reviewer #1 (Public review): https://doi.org/10.7554/eLife.106697.3.sa1
Reviewer #2 (Public review): https://doi.org/10.7554/eLife.106697.3.sa2
Author response https://doi.org/10.7554/eLife.106697.3.sa3

## Additional files

### Supplementary files
MDAR checklist

### Data availability
Data underlying the plots in *Figure 5* are available in *Figure 5—source data 1*. Crystal structures previously deposited in the PDB were used in this study: PDB accession codes 6ONI [https://doi.org/10.2210/pdb6ONI/pdb] and 8ZFS [https://doi.org/10.2210/pdb8ZFS/pdb]. Previously published NMR chemical shift assignments were used in this study: BMRB accession code 50000 [https://doi.org/10.13018/BMR50000].

The following previously published datasets were used:

| Author(s) | Year | Dataset title | Dataset URL | Database and Identifier |
|---|---|---|---|---|
| Douglas K, Jinsai S | 2020 | PPARg LBD bound to the inverse agonist T0070907 and NCoR corepressor peptide | https://doi.org/10.13018/BMR50000 | Biological Magnetic Resonance Data Bank, 10.13018/BMR50000 |
| Shang J, Kojetin DJ | 2020 | Crystal structure of PPARgamma ligand binding domain in complex with N-CoR peptide and inverse agonist T0070907 | https://doi.org/10.2210/pdb6ONI/pdb | Worldwide Protein Data Bank, 10.2210/pdb6ONI/pdb |
| Shang J, Kojetin DJ | 2024 | Crystal Structure of Human PPARgamma Ligand Binding Domain in Complex with T0070907 and MRL24 | https://doi.org/10.2210/pdb8ZFS/pdb | Worldwide Protein Data Bank, 10.2210/pdb8ZFS/pdb |

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
