## [Editor Report · eLife Assessment]

This manuscript presents a **fundamental** advance in our understanding of nuclear receptor pharmacology by expanding on previous work demonstrating dual ligand occupancy in the peroxisome proliferator-activated receptor-gamma (PPARγ). Using a **compelling** combination of biophysical, biochemical, and cellular approaches, the authors show that covalent inverse agonists with enhanced efficacy shift the receptor conformation toward a transcriptionally repressive state that limits orthosteric ligand co-binding more effectively. This revised manuscript further strengthens support for a proximal, bidirectional allosteric model of dual ligand occupancy by sharpening the distinction between prior and new findings, adding clear conceptual figures, and strengthening statistical rigor.

---

## [Referee Report · Reviewer #1 (Public review)]

Summary:

This paper focuses on understanding how covalent inhibitors of peroxisome proliferator-activated receptor-gamma (PPARg) show improved inverse agonist activities. This work is important because PPARg plays essential roles in metabolic regulation, insulin sensitization, and adipogenesis. Like other nuclear receptors, PPARg, is a ligand-responsive transcriptional regulator. Its important role, coupled with its ligand-sensitive transcriptional activities, makes it an attractive therapeutic target for diabetes, inflammation, fibrosis, and cancer. Traditional non-covalent ligands like thiazolininediones (TZDs) show clinical benefit in metabolic diseases, but utility is limited by off-target effects and transient receptor engagement. In previous studies, the authors characterized and developed covalent PPARg inhibitors with improved inverse agonist activities. They also showed that these molecules engage unique PPARg ligand binding domain (LBD) conformations whereby the c-terminal helix 12 penetrates into the orthosteric binding pocket to stabilize a repressive state. In the nuclear receptor superclass of proteins, helix 12 is an allosteric switch that governs pharmacologic responses, and this new conformation was highly novel. In this study, the authors did a more thorough analysis of how two covalent inhibitors, SR33065 and SR36708 influence the structural dynamics of PPARg LBD.

Strengths:

(1) The authors employed a compelling integrated biochemical and biophysical approach.

(2) The cobinding studies are unique for the field of nuclear receptor structural biology, and I'm not aware of any similar structural mechanism described for this class of proteins.

(3) Overall, the results support their conclusions.

(4) The results open up exciting possibilities for the development of new ligands that exploit the potential bidirectional relationship between the covalent versus non-covalent ligands studied here.

Weaknesses:

All weaknesses were addressed by the Authors in revision.

---

## [Referee Report · Reviewer #2 (Public review)]

Summary:

The authors use ligands (inverse agonists, partial agonists) for PPAR, and coactivators and corepressors, to investigate how ligands and cofactors interact in a complex manner to achieve functional outcomes (repressive vs. activating).

Strengths:

The data (mostly biophysical data) are compelling from well-designed experiments. Figures are clearly illustrated. The conclusions are supported by these compelling data. These results contribute to our fundamental understanding of the complex ligand-cofactor-receptor interactions.

Weaknesses:

Breaking down a complex system into a simpler model system, when possible, provides a unique lens with which to probe systems with mechanistic insight. While it works well in this particular paper, in general, caution should be taken when using simplified models to study a complex system.

---

## [Author Response]

The following is the authors’ response to the original reviews.

**Public Reviews:**

**Reviewer #1 (Public review):**
Summary:This paper focuses on understanding how covalent inhibitors of peroxisome proliferator-activated receptor-gamma (PPARg) show improved inverse agonist activities. This work is important because PPARg plays essential roles in metabolic regulation, insulin sensitization, and adipogenesis. Like other nuclear receptors, PPARg, is a ligand-responsive transcriptional regulator. Its important role, coupled with its ligand-sensitive transcriptional activities, makes it an attractive therapeutic target for diabetes, inflammation, fibrosis, and cancer. Traditional non-covalent ligands like thiazolininediones (TZDs) show clinical benefit in metabolic diseases, but utility is limited by off-target effects and transient receptor engagement. In previous studies, the authors characterized and developed covalent PPARg inhibitors with improved inverse agonist activities. They also showed that these molecules engage unique PPARg ligand binding domain (LBD) conformations whereby the c-terminal helix 12 penetrates into the orthosteric binding pocket to stabilize a repressive state. In the nuclear receptor superclass of proteins, helix 12 is an allosteric switch that governs pharmacologic responses, and this new conformation was highly novel. In this study, the authors did a more thorough analysis of how two covalent inhibitors, SR33065 and SR36708 influence the structural dynamics of PPARg LBD.Strengths:(1) The authors employed a compelling integrated biochemical and biophysical approach.(2) The cobinding studies are unique for the field of nuclear receptor structural biology, and I'm not aware of any similar structural mechanism described for this class of proteins.(3) Overall, the results support their conclusions.(4) The results open up exciting possibilities for the development of new ligands that exploit the potential bidirectional relationship between the covalent versus non-covalent ligands studied here.Weaknesses:(1) The major weakness in this work is that it is hard to appreciate what these shifting allosteric ensembles actually look like on the protein structure. Additional graphical representations would really help convey the exciting results of this study.

We thank the review for the comments. In response to the specific recommendations below, we added two new figures—Figure 1 and Figure 8 in this resubmission—that hopefully address the weakness identified by the reviewer.

**Reviewer #2 (Public review):**
Summary:The authors use ligands (inverse agonists, partial agonists) for PPAR, and coactivators and corepressors, to investigate how ligands and cofactors interact in a complex manner to achieve functional outcomes (repressive vs. activating).Strengths:The data (mostly biophysical data) are compelling from well-designed experiments. Figures are clearly illustrated. The conclusions are supported by these compelling data. These results contribute to our fundamental understanding of the complex ligand-cofactor-receptor interactions.Weaknesses:This is not the weakness of this particular paper, but the general limitation in using simplified models to study a complex system.

We appreciate the reviewer’s comments. Breaking down a complex system into a simpler model system, when possible, provides a unique lens with which to probe systems with mechanistic insight. While simplified models may not always explain the complexity of systems in cells, for example, our recently published work showed that a simplified model system — biochemical assays using reconstituted PPARγ ligand-binding domain (LBD) protein and peptides derived from coregulator proteins (similar to the assays in this current work) and protein NMR structural biology studies using PPARγ LBD — can explain the activity of ligand-induced PPARγ activation and repression to a high degree (pearson/spearman correlation coefficients ~0.7-0.9):

MacTavish BS, Zhu D, Shang J, Shao Q, He Y, Yang ZJ, Kamenecka TM, Kojetin DJ. Ligand efficacy shifts a nuclear receptor conformational ensemble between transcriptionally active and repressive states. Nat Commun. 2025 Feb 28;16(1):2065. doi: 10.1038/s41467-025-57325-4. PMID: 40021712; PMCID: PMC11871303.

**Recommendations for the authors**

**Reviewer #1 (Recommendations for the authors):**
(1) More set-up is needed in the results section. The first paragraph is unclear on what is new to this study versus what was done previously. Likewise, a brief description of the assays used and the meaning behind differences in signals would help the general reader along.

We modified the last paragraph of the introduction and first results section to hopefully better set the stage for what was done previously vs. what is new/recollected in this study. In our results section, we also include more description about what the assays measure.

(2) Since this paper is building on previous work, additional figures are needed in the introduction and discussion. Graphical depictions of what was found in the first study on how these ligands uniquely influence PPARg LBD conformation. A new model/depiction in the discussion for what was learned and its context with the rest of the field.

Our revised manuscript includes a new Figure 1 describing the possible allosteric mechanism by which a covalent ligand inhibits binding of other non-covalent ligands that was inferred from our previous study; and a new Figure 8 with a model for what has been learned.

(3) It is stated that the results shown are representative data for at least two biological replicates. However, I do not see the other replicates shown in the supplementary information.

We appreciate the Reviewer’s emphasis on data reproducibility and rigor. We confirm that the biochemical and cellular assay data presented are indeed representative of consistent findings observed across two or more biological replicates—and we show representative data in our figures but not the extensive replicate data in supplementary information consistent with standard practices.

(4) Figure 1a could benefit from labels of antagonists, inverse agonist, etc., next to each chemical structure. Likewise, if any co-crystal or other models are available it would be helpful to include those for comparison.

We added the pharmacological labels to Figure 2a (old Figure 1a).

(5) The figure legends don't seem to match up completely with the figures. For example, Figure 2b states that fitted Ki values +/- standard deviation. are stated in the legend, but it's shown as the log Ki.

We revised the figure legends to ensure they display the appropriate errors as reported from the data fitting.

(6) EC50, IC50, Ki, and Kd values alongside reported errors and R2 values for the fits should be reported in a table.

Our revised manuscript now includes a Source Data file (Figure 5—source data 1.xlsx) of the data (n=2) plotted in Figure 5 (old Figure 4) so that readers can regenerate the plots and calculate the errors and R2 values if desired. Otherwise, fitted values and errors are reported in figures when fitting in Prism permitted and reported errors; when Prism was unable to fit data or fit the error, n.d. (not determined) is specified.

(7) Statistical analysis is missing in some places, for example, Figure 1b.

We revised Figure 2b (old Figure 1b) to include statistical testing.

**Reviewer #2 (Recommendations for the authors):**
I suggest that the authors discuss the following points to broaden the significance of the results:(1) The two partial agonists MRL24 and nTZDpa are "partial" in the coactivator and corepressor recruitment assays, but are "complete" in the TR-FRET ligand displacement assay (Figure 2). Please explain that a partial agonist is defined based on the functional outcome (cofactor recruitment in this study) but not binding affinity/efficacy.

We added the following sentence to describe the partial agonist activity of these compounds: “These high affinity ligands are partial agonists as defined on their functional outcome in coregulator recruitment and cellular transcription; i.e., they are less efficacious than full agonists at recruiting peptides derived from coactivator proteins in biochemical assays (Chrisman et al., 2018; Shang et al., 2019; Shang and Kojetin, 2024) and increasing PPARγ-mediated transcription (Acton et al., 2005; Berger et al., 2003).“

(2) Will the discovery reported here be broadly applicable?(a) Applicable if other partial agonists and inhibitors are used?(b) Applicable if different coactivators/corepressors, or different segments of the same cofactor, are used?(c) Applicable to other NRs (their AF-2 are similar but with sequence variation)?(d) The term "allosteric" might mean different things to different people - many readers might think that it means a "distal and unrelated" binding pocket. It might be helpful to point out that in this study, the allosteric site is actually "proximal and related".

We expanded our introduction and/or discussion sections to expand upon these concepts; specific answers as follows:

(a) Orthosteric partial agonists?—yes, because helix 12 would clash with an orthosteiric ligand; other covalent inhibitors?—it depends on whether the covalent inhibitor stabilizes helix 12 in the orthosteric pocket.

(b) yes with some nuanced exceptions where certain segments of the same coregulator protein bind with high affinity and others apparently do not bind or bind with low affinity

(c) it is not clear yet if other NRs share a similar ligand-induced conformational ensemble to PPARγ

(d) we addressed this point in the 4th paragraph of the introduction “...the non-covalent ligand binding event we previously described at the alternate/allosteric site, which is proximal to the orthosteric ligand-binding pocket, …”